# Clinical Characteristics and Associated Factors for Mortality in Patients with Carbapenem-Resistant *Enterobacteriaceae* Bloodstream Infection

**DOI:** 10.3390/microorganisms11051121

**Published:** 2023-04-25

**Authors:** Jin Young Ahn, Sang Min Ahn, Jung Ho Kim, Su Jin Jeong, Nam Su Ku, Jun Yong Choi, Joon Sup Yeom, Je Eun Song

**Affiliations:** 1Department of Internal Medicine, Yonsei University College of Medicine, Seoul 03722, Republic of Korea; 2Department of Internal Medicine, Inje University Ilsan Paik Hospital, Goyang 10380, Republic of Korea

**Keywords:** carbapenem-resistant *Enterobacteriaceae*, carbapenemase-producing *Enterobacteriaceae*, bloodstream infection, mortality

## Abstract

Background: Bloodstream infection (BSI) caused by carbapenem-resistant *Enterobacteriaceae* (CRE) significantly influences patient morbidity and mortality. We aimed to identify the characteristics, outcomes, and risk factors of mortality in adult patients with CRE bacteremia and elucidate the differences between carbapenemase-producing (CP)-CRE BSI and non-CP-CRE BSI. Methods: This retrospective study included 147 patients who developed CRE BSI between January 2016 and January 2019 at a large tertiary care hospital in South Korea. The patient demographic characteristics and clinical and microbiological data including the *Enterobacteriaceae* species and carbapenemase type were obtained and analyzed. Results: *Klebsiella pneumoniae* was the most commonly detected pathogen (80.3%), followed by *Escherichia coli* (15.0%). In total, 128 (87.1%) isolates were found to express carbapenemase, and most CP-CRE isolates harbored *bla*_KPC_. The 14-day and 30-day mortality rates for CRE BSI were 34.0% and 42.2%, respectively. Higher body mass index (odds ratio (OR), 1.123; 95% confidence interval (CI), 1.012–1.246; *p* = 0.029), higher sequential organ failure assessment (SOFA) score (OR, 1.206; 95% CI, 1.073–1.356; *p* = 0.002), and previous antibiotic use (OR, 0.163; 95% CI, 0.028–0.933; *p* = 0.042) were independent risk factors for the 14-day mortality. A high SOFA score (OR, 1.208; 95% CI; 1.081–0.349; *p* = 0.001) was the only independent risk factor for 30-day mortality. Carbapenemase production and appropriate antibiotic treatment were not associated with high 14- or 30-day mortality rates. Conclusions: Mortality from CRE BSI was related to the severity of the infection rather than to carbapenemase production or antibiotic treatment, showing that efforts to prevent CRE acquisition rather than treatment following CRE BSI detection would be more effective at reducing mortality.

## 1. Introduction

*Enterobacteriaceae* are causative organisms of approximately 30% of healthcare-associated infections [1]. Recently, increased multidrug resistance of *Enterobacteriaceae* to carbapenems has become a worldwide concern [2,3,4,5]. In Korea, multidrug-resistant organisms, including *Enterobacteriaceae*, pose a tremendous clinical and economic burden [6,7]. Although the rate of carbapenem resistance in bloodstream infections (BSIs) caused by *Enterobacteriaceae* is low (1.2% for *Klebsiella pneumoniae* (KPN)), the rate of carbapenem resistance in hospital-acquired BSIs is high; this value can reach up to 4.3% in patients treated in intensive care units (ICUs) [8]. Particularly, BSIs caused by carbapenem-resistant *Enterobacteriaceae* (CRE) significantly impact patient morbidity and mortality [9,10]. A study conducted in Israel found that a longer length of hospital stay increased the rate of antimicrobial-resistant Gram-negative bacteria in BSI and was associated with higher mortality rates [11]. CRE are classified into carbapenemase-producing CRE (CP-CRE) and non-carbapenemase-producing CRE (non-CP-CRE) based on the mechanism underlying the acquisition of carbapenem resistance. However, there is a lack of studies comparing the characteristics and outcomes between CP-CRE BSI and non-CP-CRE BSI. Furthermore, there is a scarcity of information about management strategies and antibiotic treatment [12]. 

Furthermore, the resistance of CRE to most other classes of antibiotics in addition to carbapenems has severely limited treatment options. Despite attempts to use various antibiotic combinations and the development of new antibiotics to treat CRE infections, a consensus has not been reached on the optimal treatment regimen. Therefore, identifying the risk factors for mortality in patient management is imperative to identify high-risk patients. Understanding the burden of antimicrobial resistance and identifying and targeting the pathogen is important [13]. Moreover, management strategies for CRE, which include patient isolation, vary across countries based on epidemiology and resistance patterns, and research on the characteristics and prognosis of CRE-induced BSI in Korea is lacking.

Therefore, we aimed to identify the characteristics and outcomes of patients with CRE BSI and elucidate the risk factors for mortality in adult patients with CRE BSI. We also investigated whether the clinical characteristics and prognosis differ between CP-CRE BSI and non-CP-CRE BSI.

## 2. Methods

### 2.1. Study Design and Population

This was a retrospective observational cohort study performed in Seoul, Korea, at a 2500-bed tertiary care multi-specialty university hospital with a 105-bed ICU. The patients included in this study were older than 18 years (mean age 61.79 years) and hospitalized with CRE BSI between January 2016 and January 2019. In total, 147 patients were inducted in this study (36, 47, 59, and 5 in 2016, 2017, 2018, and 2019, respectively). The microbiological data from blood culture samples were obtained retrospectively from the laboratory information system records for the specified period. All the patients were included for analysis only once, at their first CRE isolation, although CRE was isolated from some patients multiple times during the study period. This study was approved by the institutional review board of the Yonsei University Health System Clinical Trials Center, and the protocol adhered to the tenets of the Declaration of Helsinki. Since the study was retrospective and the study participants were anonymized, the institutional review board waived the requirement for written consent from the patients.

### 2.2. Data Collection and Parameter Definition

The collected data included patient demographic characteristics such as age, sex, and body mass index (BMI). The clinical and microbiological variables for which data were collected included vital signs, underlying disease, route of infection, the severity of illness (classified using the sequential organ failure assessment (SOFA) score and Pitt bacteremia score), duration from admission to BSI diagnosis, previous antibiotic use, antimicrobial therapy regimen to treat the CRE infection, isolated pathogens, the presence and the type of carbapenemase of the bacterial isolate, and the antimicrobial susceptibility of the bacterial isolate. 

CRE BSI was defined as the presence of CRE in one or more blood cultures. BSI diagnosis was defined as the date of the first blood culture test that isolated CRE. Antimicrobial susceptibility tests were performed based on standard guidelines provided by the Clinical and Laboratory Standard Institute (CLSI) in M100-S25 [14]. Based on minimum inhibitory concentration (MIC) values, we defined CRE as isolates with MIC values ≥4 µg/mL for doripenem, imipenem, meropenem, and ≥2 µg/mL for ertapenem. If the Enterobacteriaceae isolate was resistant to carbapenem, we performed the modified Hodge test (MHT) and designated it as CP-CRE when the isolate with positive MHT had any carbapenemase gene in the multiplex PCR. CP-CRE were defined as CRE isolates that produced any carbapenemase including *Klebsiella pneumoniae* carbapenemase (KPC), New Delhi metallo-β-lactamase (NDM), imipenemase (IMP), GES, Verona integron-encoded metallo-β-lactamase (VIM), or oxacillinase (OXA)-48-like (OXA-48). 

Comorbidities were defined according to the International Classification of Diseases, 10th Revision. The primary focus of the BSI was classified according to the Centers for Disease Control and Prevention/National Healthcare Safety Network surveillance criteria [15]. Neutropenia was defined as an absolute neutrophil count of >500 cells/μL. Corticosteroid use was noted only if the patient had recently received 30 mg of prednisone daily for at least 7 days or 20 mg/day for 14 days in the previous 30 days. Receipt of immunosuppressants was defined as the use of any immunosuppressive drug (e.g., cyclosporine or antineoplastic chemotherapy) in the previous 30 days. Previous exposure to various antibiotic agents was also noted. Exposure to a specific antimicrobial agent was considered significant in our analysis only if it had been administered for at least three consecutive days within a month before the CRE BSI onset. Antimicrobial therapy was considered appropriate if the causative microorganisms were susceptible to at least one of the administered antimicrobial agents after the identification of Gram-negative organisms in the blood culture. Exposure to various risk factors was taken into consideration for the analysis only if it had occurred before the development of the CRE BSI.

### 2.3. Statistical Analysis

The primary and secondary endpoints were all-cause 14- and 30-day mortality rates, respectively. SPSS (version 25.0; IBM Corporation, Armonk, NY, USA) was used for all the statistical analyses. Categorical variables were analyzed using the chi-squared test or Fisher’s exact test. Continuous variables were analyzed using an independent samples t-test or a Mann–Whitney U test. Logistic regression analysis was used to assess the effect of independent variables on the risk. A two-tailed *p*-value <0.05 was considered significant. All variables with *p*-values <0.05 in the univariate analysis were included in the multiple logistic regression models that were used to identify the risk factors for mortality after the CRE BSI onset.

## 3. Results

### 3.1. Clinical Characteristics of CRE Bloodstream Infections

The demographic and clinical characteristics of the patients with CRE BSI are shown in Table 1. The mean age was 61.79 years and 60.5% (89/147) of patients were male. The median duration from admission to BSI diagnosis was 18 days (interquartile range (IQR), 5–57 days), and 42% (46/147) of the patients had a condition requiring their admission to the ICU at the time of CRE BSI diagnosis. Most CRE BSI cases were due to CP-CRE (87.1%, 128/147); however, 12.9% (19/147) were due to non-CP-CRE. Pneumonia (31.3%) and catheter-related BSI (27.9%) were the most common sources of infection. The mean SOFA score at the time of BSI diagnosis was 5.74 ± 4.31, and the mean Pitt bacteremia score was 2.19 ± 2.27. Most patients (83%) were previously exposed to broad-spectrum antibiotics and 45.6% had a history of carbapenem use. KPN was the most common pathogen (80.3%, 118/147), followed by *Escherichia coli* (15.0%, 22/147) and *Enterobacter* sp. (6.1%, 9/147). Most of the CP-CRE isolates harbored *bla*_KPC_ (87.5%, 112/128), followed by *bla*_NDM_ (10.9%, 14/128). The 14-day and 30-day mortality rates for the CRE BSI were 34.0% (50/147) and 42.2% (62/147), respectively.

### 3.2. CP-CRE and Non-CP-CRE Bloodstream Infections

The baseline characteristics of the patients such as age, sex, and BMI did not differ significantly between the CP-CRE BSI and non-CP-CRE BSI groups (Table 1). Similarly, the time of BSI diagnosis during hospitalization, predisposing factors, or severity of BSI did not significantly differ between the two groups. However, catheter-related BSIs were significantly more common in the non-CP-CRE BSI group than in the CP-CRE BSI group (63.2%, 12/19 vs. 22.7%, 29/128) (*p* < 0.001). Pneumonia tended to be more common in the CP-CRE BSI group (although the difference was not significant), and the number of patients with underlying chronic lung disease was significantly higher in the CP-CRE BSI group (*p* = 0.025). The number of patients with prior carbapenem use was high in the CP-CRE BSI group (*p* = 0.08), although the difference was not statistically significant. The proportion of BSIs due to KPN was significantly high in the CP-CRE BSI group (85.2% (109/128) vs. 47.4% (9/19)) (*p* < 0.001). 

Table 2 shows the distribution of the MICs of carbapenems and aminoglycosides for the isolated CRE. The proportion of CRE isolates with ertapenem MIC ≥8 µg/mL and meropenem/imipenem MIC ≥16 µg/mL was significantly higher in the CP-CRE BSI group than in the non-CP-CRE BSI group (91.2% vs. 57.9%, *p* < 0.001 for ertapenem and 83.1% vs. 7.7%, *p* < 0.001 for meropenem/imipenem). In the non-CP-CRE BSI group, 61.5% of the isolates were susceptible (MIC ≤ 1) to meropenem or imipenem, and 30.8% were not susceptible but had a meropenem/imipenem MIC <8 µg/mL, which was lower than that observed in the CP-CRE BSI group. The CP-CRE BSI group also had a significantly higher proportion of gentamicin-resistant isolates compared to the non-CP-CRE BSI group (75.0% vs. 16.7%, *p* < 0.001).

### 3.3. Associated Factors for 14- and 30-Day Mortality in CRE Bloodstream Infections

The factors associated with mortality in CRE BSI are presented in Table 3 and Table 4. Univariate analysis showed that patients who died within 14 days of CRE BSI diagnosis had a higher BMI (*p* = 0.003), higher rate of prophylactic antibiotic use (*p* = 0.020), higher presence of shock (*p* < 0.001), acute kidney injury (*p* = 0.012), higher SOFA score (*p* < 0.001), higher Pitt bacteremia score (*p* < 0.001), and a higher rate of previous antibiotic use (*p* = 0.037) than the survivors did (Table 3). The mean BMI was 22.15 ± 4.48 kg/m^2^, but the number of patients with a BMI > 25 kg/m^2^ was higher among the non-survivors than among the survivors (38.0% vs. 17.5%) (*p* = 0.006). Furthermore, 9 out of the 10 patients with a BMI > 30 kg/m^2^ were in the non-survivor group (*p* < 0.001). Multivariate logistic regression analysis showed that higher BMI (odds ratio (OR), 1.123; 95% CI, 1.012–1.246; *p* = 0.029), higher SOFA score (OR, 1.206; 95% CI, 1.073–1.356; *p* = 0.002), and previous antibiotic use (OR, 0.163; 95% CI, 0.028–0.933; *p* = 0.042) were independently associated with the 14-day mortality in CRE BSI (Table 3).

Univariate analysis showed that patients who died within 30 days of a CRE BSI diagnosis had higher BMI (*p* = 0.003), higher rates of presence of shock (*p* < 0.001) and acute kidney injury (*p* = 0.002), higher SOFA score (*p* < 0.001), higher Pitt bacteremia score (*p* < 0.001), and a higher rate of previous antibiotic use (*p* =0.043) and were more likely to have a urinary tract infection as a source of infection (*p* = 0.026) compared to the survivors (Table 3). In contrast, multivariate logistic regression analysis showed that only a higher SOFA score (OR, 1.208; 95% CI, 1.081–1.349; *p* = 0.001) was an independent risk factor for 30-day mortality (Table 4). Interestingly, carbapenemase production, carbapenemase genotype, isolated pathogen, use of specific antibiotics for CRE, and appropriate antibiotic use did not show statistically significant correlations with the 14- or 30-day mortality.

## 4. Discussion

Carbapenem-resistant Gram-negative organisms, including CRE, CR *Pseudomonas aeruginosa* (CRPA), and CR *Acinetobacter baumannii* (CRAB), pose a major global threat in the 21st century [16,17]. A 2021 study by Hovan et al. involving 155 patients with CRE BSI showed that KPN accounted for the highest proportion of cases (46.6%, 69/146) with KPC being the predominant carbapenemase in the CP-CRE group (92%, 81/88) [18]. According to a nationwide epidemiological study in Korea, the most frequent causative agents of BSI was KPN (69%), followed by *Enterobacter* sp. (10%) and *E. coli* (8%) [14]. Non-CP-CRE tends to occur sporadically owing to antibiotic selection pressure, whereas the plasmid-associated horizontal spread of CP-CRE within hospitals has also been reported [19,20,21]. In this study, most of the CP-CRE BSI cases were caused by KPC-producing KPN (85.2%, 109/128), whereas KPN accounted for 47% (9/19) of the non-CP-CRE BSI cases. This suggests that most CRE BSI cases in the hospital where this study was conducted may have been caused by the in-hospital transmission of CPE, leading to colonization. Further studies are required to confirm this conclusion including whole-genome sequencing of the CPE strains to determine the clonality. Furthermore, various diagnostic methods are being investigated for the rapid diagnosis of multidrug-resistant organisms, which plays an important role in preventing transmission [22,23,24,25].

A meta-analysis by Falagas et al. showed that the mortality attributed to CRE infection varied from 26 to 44%, and the mortality rate was higher in CRE-infected patients than in carbapenem-susceptible *Enterobacteriaceae* (CSE)-infected patients (relative ratio (RR), 2.05, 95% confidence interval (CI), 1.56–2.69) [26]. A case-control study by Zarkotou et al. showed that infection mortality was 34%. Older age, APACHE II score at infection onset, and inappropriate antibiotic treatment were independent predictors of mortality. [27]. In this study, the 30-day mortality rate was >40% and was significantly related to the severity of infection at the time of the BSI (measured using the SOFA score and Pitt bacteremia score), and mortality. However, the 14-day and 30-day mortality rates were not significantly associated with appropriate antibiotic use. Of the 50 patients who died within 14 days, 34% (17/40) died within 3 days because of severe sepsis, making it difficult to assess the effectiveness of antibiotics in these cases. Nevertheless, no significant difference was observed between the survivors and non-survivors with regard to the use of amikacin, colistin, or combination therapy with colistin and carbapenem. The percentage of broad-spectrum antibiotics administered before the onset of infection was significantly higher in the mortality group. This suggests that in critically ill patients with a long history of broad-spectrum antibiotic use, CRE BSI developed into a breakthrough BSI that could not be improved with susceptible antibiotics alone. The presence of difficult-to-treat resistance, defined as resistance to all first-line antibiotics in Gram-negative BSI, is associated with decreased survival rates because of the challenge of finding an effective antibiotic to treat it [28]. Clinicians are prescribing a wide range of treatment regimens in their current practice with no proven efficacy and potential side effects, as available studies do not provide clear recommendations on the type and number of agents to be used in combination [29]. Furthermore, the recommended antibiotics for treating KPC-producing CP-CRE BSI such as ceftazidime-avibactam or cefiderocol [30,31] are not available in Korea. In another Korean study dealing with CRPA and CRAB BSI, carbapenem resistance was an independent risk factor for treatment failure regardless of the appropriateness of empiric antibiotics [32]. Caution is needed when Korean national studies conclude that appropriate antibiotic use is not associated with reduced mortality. 

In this study, there was no significant difference between the CP-CRE group and the non-CP-CRE group, except for catheter-related BSIs as the source of infection and underlying lung disease. The non-CP-CRE group showed a diverse range of strains, suggesting that there was no clonal spread and that the infection occurred sporadically, unlike the CP-CRE group. In this study, we observed a significant difference between the groups in the MIC values of carbapenem, with the CP-CRE BSI group showing higher MIC values. These findings are consistent with that of a study by Tamma et al., which reported that CP-CRE pathogens are more likely to have meropenem MICs of ≥16 μg/mL than non-CP-CRE isolates (38% vs. 2%), thus contributing to higher mortality in the CP-CRE BSI group compared to that of the non-CP-CRE BSI group [33]. In a study of CRPA infections by Reyes et al., patients with CP-CRPA infections had higher 30-day mortality than those with non-CP-CRPA infections [34]. However, in this study, we did not observe any differences in mortality between the patients with CP-CRE BSI and non-CP-CRE BSI. This finding is consistent with the fact that the use of susceptible antibiotics at this hospital was not significantly associated with mortality. A Korean study in 2020 also found no significant difference in the 14-day mortality between CP-CRE BSI and non-CP-CRE BSI. These conflicting results highlight the need for further research to develop effective treatment strategies for CRE infection.

In our study, higher BMI was an independent risk factor for 14-day mortality. Obesity has been associated with an increased risk of nosocomial infection and sepsis events [35,36]. However, information on the impact of obesity on the outcomes of patients with sepsis is lacking. Obese patients may have insufficient blood antibiotic concentrations or fluid resuscitation compared to patients with a normal BMI [37]. In our study, the mean BMI was < 25 kg/m^2^ but the number of patients with BMI > 25 kg/m^2^ was higher among the non-survivors than among the survivors (38% vs. 17.5%) (*p* = 0.006). Moreover, 90% (1/9) of the patients with a BMI > 30 kg/m^2^ were in the non-survivor group (*p* < 0.001).

This study has several limitations, such as the relatively short study period, a limited number of CRE BSI cases, and retrospective design, which may have introduced bias or confounding factors. One of the fundamental limitations of retrospective studies is the inability to accurately determine whether the identified microorganisms are true pathogens. To minimize this bias, our study focused only on BSIs. Additionally, we did not perform a phylogenetic analysis to identify the clonality of the CP-CRE isolates in this study. Despite these limitations, we have analyzed the clinical characteristics and prognostic differences between the CP-CRE and non-CP-CRE isolates, focusing on clinically significant infections that cause BSI. Furthermore, the diversity of the patient population contributes to the generalizability of our findings. Additionally, we have emphasized the importance of infection control measures to prevent CRE acquisition and the need for further research to identify effective prevention strategies. Given the high mortality rate of CRE BSI and the difficulty in improving prognosis through antibiotic treatment alone, our findings provide valuable information for clinical management.

In conclusion, most of the CRE BSIs reported in this study were due to CP-CRE. However, the mortality due to CRE BSIs was not significantly associated with carbapenemase production or the antibiotic treatment regimen but rather with the severity of the infection. Therefore, infection control efforts to prevent CRE acquisition would be a more cost-effective approach to reducing patient mortality.

## Figures and Tables

**Table 1 microorganisms-11-01121-t001:** Clinical characteristics of patients with carbapenemase-producing carbapenem-resistant Enterobacteriaceae (CP-CRE) and non-CP-CRE bloodstream infections.

Characteristics	Total (*n* = 147)	Non-CP-CPE (*n* = 19)	CP-CRE (*n* = 128)	*p*-Value
Age	61.79 ± 12.73	61.42 ± 8.88	61.84 ± 13.23	0.893
Age > 65	58 (39.5)	8 (42.1)	50 (39.1)	0.806
BMI	22.15 ± 4.48	23.32 ± 5.11	21.97 ± 4.37	0.221
Male	89 (60.5)	14 (73.7)	75 (58.6)	0.314
Admission to bacteremia (day) (median, IQR)	18 (5–57)	67 (1.5–15.5)	11 (6–60.75)	0.436
Infection source				
UTI	14 (9.5)	2 (10.5)	12 (9.4)	0.873
Pneumonia	46 (31.3)	2 (10.5)	44 (34.4)	0.060
SSTI	7 (4.8)	0 (0)	7 (5.5)	0.595
CRBSI	41 (27.9)	12 (63.2)	29 (22.7)	<0.001
IAI	26 (17.7)	2 (10.5)	24 (18.8)	0.528
Underlying comorbidity				
HTN	60 (40.8)	8 (42.1)	52 (40.6)	1.000
DM	52 (35.4)	6 (31.6)	46 (35.9)	0.801
Cardiovascular disease	42 (28.6)	7 (36.8)	35 (27.3)	0.420
Lung disease	40 (27.2)	1 (5.3)	39 (30.5)	0.025
Renal disease	29 (19.7)	3 (15.8)	26 (20.3)	0.767
Solid cancer	56 (38.1)	11 (57.9)	45 (35.2)	0.076
Hematologic disease	22 (15.0)	3 (15.8)	19 (14.8)	1.000
SOT	34 (23.1)	4 (21.1)	30 (23.4)	1.000
HSCT	8 (5.4)	1 (5.3)	7 (5.5)	1.000
Rheumatologic disease	9 (6.1)	2 (10.5)	7 (5.5)	0.605
Liver disease	22 (15.0)	1 (5.3)	21 (16.4)	0.309
Predisposing factor				
Neutropenia	31 (21.1)	3 (15.8)	28 (21.9)	0.578
Steroid	45 (30.6)	5 (26.3)	40 (31.3)	0.793
Oral prophylactic antibiotics	27 (18.4)	3 (15.8)	24 (18.8)	1.000
Immunosuppressant	47 (32.0)	7 (36.8)	40 (31.3)	0.793
Chemotherapy	34 (23.1)	4 (21.1)	30 (23.4)	1.000
Pressure sore	19 (12.9)	0	19 (14.8)	0.134
Operation	32 (21.8)	3 (15.8)	29 (22.7)	0.571
Previous broad-spectrum antibiotics	122 (83.0)	15 (78.9)	107 (83.6)	0.743
Carbapenem use	67 (45.6)	5 (26.3)	62 (48.4)	0.086
Procedures				
Foley catheter	97 (66.0)	10 (52.6)	87 (68.0)	0.188
Central venous catheter	111 (75.5)	14 (73.7)	97 (75.8)	0.843
Ventilator care	38 (25.9)	4 (21.1)	34 (26.6)	0.609
Drainage catheter	57 (38.8)	6 (31.6)	51 (39.8)	0.490
Severity				
ICU care	62 (42.2)	7 (36.8)	55 (43.0)	0.632
ICU duration (days) (median, IQR)	21.5 (6.25–31.5)	23.5 (11.5–41.5)	20.5 (5.5–28)	0.916
Shock	63 (42.9)	8 (42.1)	55 (43.0)	1.000
Acute kidney injury	41 (27.9)	5 (26.3)	36 (28.1)	1.000
SOFA score	5.74 ± 4.31	5.21 ± 3.90	5.82 ± 4.38	0.567
Pitt bacteremia score	2.19 ± 2.27	2.47 ± 2.89	2.13 ± 2.15	0.549
Pathogen				
*Klebsiella pneumoniae*	118 (80.3)	9 (47.4)	109 (85.2)	<0.001
*Escherichiae coli*	22 (15.0)	3 (15.8)	19 (14.8)	1.000
*Enterobacter* spp.	9 (6.1)	7 (36.8)	2 (1.6)	<0.001
Carbapenemase				
KPC			112 (87.5)	
OXA			1 (0.8)	
NDM			14 (10.9)	
Outcomes				
Mortality (14 days)	50 (34.0)	8 (42.1)	42 (32.8)	0.445
Mortality (30 days)	62 (42.2)	11 (57.9)	51 (39.8)	0.212

BMI: body mass index; UTI: urinary tract infection; SSTI: skin and soft tissue infection; CRBSI: catheter-related bloodstream infection; IAI: intra-abdominal infection; IQR: interquartile range; HTN: hypertension; DM: diabetes mellitus; SOT: solid organ transplantation; HSCT: hematopoietic stem cell transplantation; ICU: intensive care unit; SOFA: sequential organ failure assessment.

**Table 2 microorganisms-11-01121-t002:** Differences in the distributions of the minimum inhibitory concentration values between carbapenemase-producing carbapenem-resistant *Enterobacteriaceae* (CP-CRE) and non-CP-CRE bloodstream infections.

Antimicrobial Agent	Number (%)		*p*-Value
	CP-CRE BSI Group (*n* = 128)	Non-CP-CRE BSI Group (*n* = 19)	
Ertapenem			<0.001
MIC ≤ 0.5	1 (0.8)		
0.5 < MIC <2		1 (5.3)	
2 ≤ MIC < 4	1 (0.8)	2 (10.5)	
4 ≤ MIC < 8	9 (7.2)	5 (26.3)	
MIC ≥ 8	114 (91.2)	11 (57.9)	
Meropenem/Imipenem			<0.001
MIC ≤ 1	2 (1.6)	8 (61.5)	
1 < MIC < 4	2 (1.6)	1 (7.7)	
4 ≤ MIC < 8	2 (1.6)	3 (23.1)	
8 ≤ MIC < 16	15 (12.1)		
MIC ≥ 16	103 (83.1)	1 (7.7)	
AMK → Amikacin			0.196
S (MIC ≤ 16)	121 (94.5)	16 (88.9)	
I (16 < MIC < 64)	3 (2.3)		
R (MIC ≥ 64)	4 (3.1)	2 (11.1)	
Gentamicin			<0.001
S (MIC ≤ 4)	31 (24.2)	14 (77.8)	
I (4 < MIC < 16)	1 (0.8)	1 (5.6)	
R (MIC ≥ 16)	96 (75.0)	3 (16.7)	

BSI: bloodstream infection; MIC: minimum inhibitory concentration.

**Table 3 microorganisms-11-01121-t003:** Clinical characteristics of the survivors and non-survivors with carbapenem-resistant *Enterobacteriaceae* bacteremia according to 14-day and 30-day mortality.

	14-Day Mortality			30-Day Mortality		
	Survivors (*n* = 97)	Mortality (*n* = 50)	*p*-Value	Survivors (*n* = 85)	Mortality (*n* = 62)	*p*-Value
Age	60.98 ± 12.50	63.36 ± 13.15	0.458	60.53 ± 13.14	63.52 ± 12.03	0.736
Male	60 (61.9)	29 (58.0)	0.650	49 (57.6)	40 (64.5)	0.495
BMI	21.23 ± 3.68	23.41 ± 5.15	0.003	21.23 ± 3.68	23.41 ± 5.15	0.003
Admission to bacteremia (median, IQR)	3.5 (3–56)	11 (7.5–59)	0.841	33.5 (3–52)	3.5 (7–68.5)	0.841
CPE	86 (88.7)	42 (84.0)	0.425	77 (90.6)	51 (82.3)	0.137
Infection source						
UTI	12 (12.4)	2 (4.0)	0.101	12 (14.1)	2 (3.2)	0.026
Pneumonia	29 (29.9)	17 (34.0)	0.611	25 (29.4)	21 (33.9)	0.565
SSTI	4 (4.1)	3 (6.0)	0.613	3 (3.5)	4 (6.5)	0.411
CRBSI	26 (27.1)	15 (30.0)	0.682	23 (27.1)	18 (29.0)	0.792
IAI	16 (16.5)	10 (20.0)	0.598	13 (15.3)	13 (21.0)	0.373
Underlying comorbidity						
HTN	39 (40.2)	21 (42.0)	0.834	32 (37.6)	28 (45.2)	0.360
DM	38 (39.2)	14 (28.0)	0.179	29 (34.1)	23 (37.1)	0.709
Cardiovascular disease	23 (23.7)	19 (38.0)	0.069	19 (22.4)	23 (37.1)	0.051
Chronic lung disease	26 (26.8)	14 (28.0)	0.877	24 (28.2)	16 (25.8)	0.744
Chronic renal disease	20 (20.6)	9 (18.0)	0.705	17 (20.0)	12 (19.4)	0.923
Solid cancer	41 (42.3)	15 (30.0)	0.147	33 (38.8)	23 (37.1)	0.831
Hematologic disease	12 (12.4)	10 (20.0)	0.219	12 (54.5)	10 (16.1)	0.736
HSCT	5 (5.2)	3 (6.0)	0.831	5 (5.9)	3 (4.8)	0.783
SOT	24 (24.7)	10 (20.0)	0.518	23 (27.1)	11 (17.7)	0.186
Rheumatologic disease	5 (5.2)	4 (8.0)	0.495	4 (4.7)	5 (8.1)	0.402
Chronic liver disease	14 (14.4)	8 (16.0)	0.801	13 (15.3)	9 (14.5)	0.896
Predisposing conditions						
Neutropenia	21 (21.6)	10 (20.0)	0.816	20 (23.5)	11 (17.7)	0.396
Steroid use	29 (29.9)	16 (32.0)	0.793	28 (32.9)	17 (27.4)	0.473
Prophylactic antibiotics	22 (22.7)	5 (10.0)	0.060	21 (24.7)	6 (9.7)	0.020
Immunosuppressant	33 (34.0)	14 (28.0)	0.458	31 (36.5)	16 (25.8)	0.171
Chemotherapy	26 (26.8)	8 (16.0)	0.141	22 (25.9)	12 (19.4)	0.354
Pressure sore	13 (13.4)	6 (12.0)	0.810	12 (14.1)	7 (11.3)	0.614
Surgery	24 (24.7)	8 (16.0)	0.224	22 (25.9)	10 (16.1)	0.157
Severity						
ICU care	38 (39.2)	24 (48.0)	0.305	31 (36.5)	31 (50.0)	0.101
ICU duration	8.75 ± 17.22	14.66 ± 38.74	0.203	8.52 ± 16.96	13.84 ± 35.75	0.232
Shock	28 (28.9)	35 (70.0)	<0.001	25 (29.4)	38 (61.3)	<0.001
Acute kidney injury	19 (19.6)	22 (44.0)	0.002	17 (20.0)	24 (38.7)	0.012
SOFA score	4.63 ± 4.07	7.90 ± 3.98	<0.001	4.47 ± 3.99	7.48 ± 4.15	<0.001
Pitt bacteremia score	1.60 ± 1.87	3.25 ± 2.55	<0.001	1.53 ± 1.89	3.04 ± 2.44	<0.001
Previous antibiotics use	76 (78.4)	46 (92.0)	0.037	66 (77.6)	56 (90.3)	0.043
Carbapenem	43 (44.3)	24 (48.0)	0.672	36 (42.4)	31 (50.0)	0.358
Quinolone	19 (19.6)	14 (28.0)	0.247	16 (18.8)	17 (27.4)	0.217
Cephalosporin	28 (28.9)	16 (32.0)	0.694	25 (29.4)	19 (30.6)	0.872
Pathogen						
*Klebsiella pneumoniae*	78 (80.4)	40 (80.0)	0.953	68 (80.0)	50 (80.6)	0.923
*Escherichia coli*	15 (15.5)	7 (14.0)	0.814	13 (15.3)	9 (14.5)	0.896
*Enterobacter* spp.	6 (6.2)	3 (6.0)	0.965	6 (7.1)	3 (4.8)	0.579
KPC genotype	76 (78.4)	36 (72.0)	0.392	67 (78.8)	45 (72.6)	0.380
Antibiotics treatment						
Appropriate antibiotic use	73 (76.0)	25 (75.8)	0.974	64 (76.2)	34 (75.6)	0.936
Amikacin in S	51 (53.1)	16 (48.5)	0.645	45 (53.6)	22 (48.9)	0.612
Carbapenem	73 (76.0)	24 (72.7)	0.704	64 (76.2)	33 (73.3)	0.720
Carbapenem (prolonged infusion)	10 (10.4)	3 (9.1)	0.827	10 (11.9)	3 (6.7)	0.346
Tigecycline	4 (4.2)	4 (12.1)	0.102	4 (4.8)	4 (8.9)	0.354
Colistin	32 (33.3)	13 (39.4)	0.529	28 (33.3)	17 (37.8)	0.614
Carbapenem/colistin combination	27 (28.1)	12 (36.4)	0.374	24 (28.6)	15 (33.3)	0.575

BMI: body mass index; CPE: carbapenemase-producing *Enterobacteriaceae*; UTI: urinary tract infection; SSTI: skin and soft tissue infection; CRBSI: catheter-related bloodstream infection; IAI: intra-abdominal infection; IQR: interquartile range; HTN: hypertension; DM: diabetes mellitus; SOT: solid organ transplantation; HSCT: hematopoietic stem cell transplantation; ICU: intensive care unit; SOFA: sequential organ failure assessment.

**Table 4 microorganisms-11-01121-t004:** Multivariate analysis of the risk factors of 14-day and 30-day mortality due to carbapenem-resistant Enterobacteriaceae bacteremia.

Characteristics	Univariate Analysis		Multivariate Analysis	
	Unadjusted OR (95% CI)	*p*-Value	Adjusted OR (95% CI)	*p*-Value
14-day mortality				
Age	1.015 (0.987–1.044)	0.283	1.037 (0.993–1.083)	0.105
BMI	1.128 (1.039–1.225)	0.04	1.123 (1.012–1.246)	0.029
CPE	0.672 (0.251–1.794)	0.427	1.176 (0.297–4.651)	0.817
SOFA score	1.201 (1.100–1.311)	<0.001	1.206 (1.073–1.356)	0.002
Previous antibiotics use	3.178 (1.026–9.839)	0.045	0.163 (0.028–0.933)	0.042
Amikacin treatment	0.830 (0.376–1.833)	0.646	1.522 (0.573–4.041)	0.399
Carbapenem treatment	0.840 (0.342–2.063)	0.84	1.595 (0.534–4.760)	0.403
Colistin treatment	1.378 (0.616–3.081)	0.435	1.187 (0.432–3.266)	0.739
30-day mortality				
Age	1.019 (0.992–1.047)	0.162	1.037 (0.998–1.079)	0.066
BMI	1.123 (1.035–1.217)	0.005	1.095 (0.994–1.207)	0.067
CPE	0.482 (0.181–1.280)	0.143	0.505 (0.144–1.770)	0.286
Urinary tract infection	0.203 (0.044–0.942)	0.026	0.270 (0.031–2.393)	0.240
SOFA score	1.192 (1.093–1.301)	<0.001	1.194 (1.068–1.335)	0.002
Previous antibiotics use	2.687 (1.004–7.191)	0.049	3.005 (0.739–12.214)	0.124
Amikacin treatment	0.829 (0.402–1.711)	0.612	0.654 (0.267–1.604)	0.354
Carbapenem treatment	0.859 (0.375–1.970)	0.72	0.760 (0.269–2.145)	0.604
Colistin treatment	1.263 (0.598–2.665)	0.540	0.812 (0.313–2.104)	0.668

BMI: body mass index; CPE: carbapenemase-producing *Enterobacteriaceae*; SOFA: sequential organ failure assessment; OR: odds ratio; CI: confidence interval.

## Data Availability

Data will be available upon justified request.

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
