# Peer review of "Clinical Characteristics and Associated Factors for Mortality in Patients with Carbapenem-Resistant Enterobacteriaceae Bloodstream Infection"

_microorganisms, 2023, doi:10.3390/microorganisms11051121_

Round 1

Reviewer 1 Report

The manuscript is well written and the content is well explained. The tables contain a lot of information and the text is well synthesized. The major limitation, as indicated by the authors, is the small number of isolates. There is nothing new in the manuscript, but it reflects the data on this particular country. 

Minor comments:

Authors should rewrite the italics of the species.

Author Response

Point 1: The manuscript is well written and the content is well explained. The tables contain a lot of information and the text is well synthesized. The major limitation, as indicated by the authors, is the small number of isolates. There is nothing new in the manuscript, but it reflects the data on this particular country. 

Minor comments: Authors should rewrite the italics of the species.

Response 1: We have carefully reviewed the manuscript and corrected the italicized species as suggested by the reviewer.

**We have had the manuscript professionally proofread to ensure that it meets the highest standards of English usage. We are confident that the revised version has addressed theses issues and will be more readable to readers.

We would also like to mention that all the revisions and changes made to the manuscript have been clearly marked and indicated to make it easier for the reviewers to identify. We have used a different font color to distinguish the revised portions from the original text.

Reviewer 2 Report

This manuscript identified the characteristics and risk factors of mortality in adult patients with CRE bacteremia. This study was well designed and conducted. The data presented in the study is interesting to readers, but there are some concerns:

1.      The 14-day mortality and the 30-day mortality need more explanation because it is not clear if the deaths were directly caused by the infection or not. If the deaths were not caused by the infection, it is hard to make any conclusions with the current data.

2.      The manuscript will be more interesting if the authors discuss the reasons for the differences between CP-CRE BSI and non-CP-CRE BSI groups.

3.      There are numerous typos and grammar mistakes in the manuscripts.

This manuscript identified the characteristics and risk factors of mortality in adult patients with CRE bacteremia. This study was well designed and conducted. The data presented in the study is interesting to readers, but there are some concerns:

1.      The 14-day mortality and the 30-day mortality need more explanation because it is not clear if the deaths were directly caused by the infection or not. If the deaths were not caused by the infection, it is hard to make any conclusions with the current data.

2.      The manuscript will be more interesting if the authors discuss the reasons for the differences between CP-CRE BSI and non-CP-CRE BSI groups.

3.      There are numerous typos and grammar mistakes in the manuscripts.

Author Response

------------------------------------------------------------

This manuscript identified the characteristics and risk factors of mortality in adult patients with CRE bacteremia. This study was well designed and conducted. The data presented in the study is interesting to readers, but there are some concerns:

Point 1:  The 14-day mortality and the 30-day mortality need more explanation because it is not clear if the deaths were directly caused by the infection or not. If the deaths were not caused by the infection, it is hard to make any conclusions with the current data.

Response 1: We have added a sentence to the limitation part of the discussion section, stating that it is a fundamental limitation of retrospective studies to not be able to prove the exact correlation between the presence of certain bacterial species and their pathogenicity. Furthermore, in order to minimize this bias, we focused solely on BSI in our study.

  • One of the fundamental limitations of retrospective studies is the inability to accurately determine whether the identified microorganisms are true pathogens. To minimize this bias, our study focused only on BSIs.

Point 2:  The manuscript will be more interesting if the authors discuss the reasons for the differences between CP-CRE BSI and non-CP-CRE BSI groups.

Response 2: We added several sentences in the discussion section about the differences between CP-CRE BSI and non-CP-CRE BSI groups.

  • In this study, there was no significant difference between the CP-CRE group and the non-CP-CRE group, except for CRBSI as the source of infection and underlying lung disease. The non-CP-CRE group showed a diverse range of strains, suggesting that there was no clonal spread that the infection occurred sporadically, unlike the CP-CRE group.

Point 3:  There are numerous typos and grammar mistakes in the manuscripts.

Response 3: We have had the manuscript professionally proofread to ensure that it meets the highest standards of English usage. We are confident that the revised version has addressed theses issues and will be more readable to readers.

We would also like to mention that all the revisions and changes made to the manuscript have been clearly marked and indicated to make it easier for the reviewers to identify. We have used a different font color to distinguish the revised portions from the original text.

**We include an English proofreading certificate below.

Reviewer 3 Report

In the introduction, the following references should be commented on:

Lancet 2022; 399:629.

Jean SS. Front Cell Infect Microbiol 2022; 12:823684.

Nutman A. Microorganisms 2022; 10:1009.

Zhang Z. Infect Drug Resist 2022; 15:249.

Marimuthu K Clin Infect Dis 2017; 64:s68.

A discussion of the laboratory methods for the detection of a carbapenemase-producing bacteria should be included:

Ong SWX Clin Infect Dis 2022; 74:1859.

Gu D. Antibiotics (Basel) 2023; 12:300.

Gill CM. Emerg Microbes Infect 2023; 12:2179344.

Camalena F. Microbiol Spectr 2023; 11:e0254722.

Other carbapenemase-producing bacteria should be commented on:

Acinetobacter baumannii:

Falcone M. Clin Infect Dis 2023.

Pogue JM. BMC Infect Dis 2022; 22:36.

Pseudomonas aeruginosa:

Lee CM. Sci Rep 2022; 12:8527.

Reyes J. Lancet Microbe 2023; 4:e159.

In discussion comments on treatment:

Savoldi A. BMC Infect Dis 2021; 21:545.

Paul M. Clin Microbiol Infect 2022; 28:521.

Kadri SS. Clin Infect Dis 2018; 67:1803.

In conclusion comment on:

Song KH. J Glob Antimicrob Resist 2022; 31:379.

Some typos in introduction: Klebsiella pneumonia should be Klebsiella pneumoniae.

None

Author Response

-----------------------------------------------------------------------

Point 1: In the introduction, the following references should be commented on:

Lancet 2022; 399:629.

Jean SS. Front Cell Infect Microbiol 2022; 12:823684

Nutman A. Microorganisms 2022; 10:1009.

Zhang Z. Infect Drug Resist 2022; 15:249.

Marimuthu K Clin Infect Dis 2017; 64:s68.

Response 1: Following the reveiwer’s recommendation, we have inserted the references as references [12], [3], [11], [4], and [5], respectively. Based on these references, we have added the following sentences to the introduction section.

  • Understanding the burden of antimicrobial resistance and identifying and targeting the pathogen is important [12].
  • Recently, increased multidrug resistance of Enterobacteriaceae to carbapenems has become a worldwide concern [2-5].
  • A study conducted in Israel found that longer length of hospital stay increased the rate of antimicrobial-resistant gram-negative bacteria in BSI and was associated with higher mortality rates [11].

Point 2: A discussion of the laboratory methods for the detection of a carbapenemase-producing bacteria should be included:

Ong SWX Clin Infect Dis 2022; 74:1859.

Gu D. Antibiotics (Basel) 2023; 12:300.

Gill CM. Emerg Microbes Infect 2023; 12:2179344.

Camalena F. Microbiol Spectr 2023; 11:e0254722.

Response 2: Following the reveiwer’s recommendation, we have inserted the references as references [24], [23], [22], and [21], respectively. Based on these references, we added these references by mentioning the detection of a carbapenemase-producing bacterium in the discussion section.

  • Furthermore, various diagnostic methods are being investigated for the rapid diagnosis of multidrug-resistant organisms, which plays an important role in preventing transmission [21-24].

Point 3: Other carbapenemase-producing bacteria should be commented on:

Acinetobacter baumannii:

Falcone M. Clin Infect Dis 2023.

Pogue JM. BMC Infect Dis 2022; 22:36.

Pseudomonas aeruginosa:

Lee CM. Sci Rep 2022; 12:8527.

Reyes J. Lancet Microbe 2023; 4:e159.

Response 3: Following the reveiwer’s recommendation, we have inserted the references as references [15], [16], [31], and [33], respectively. Based on these references, we have added the following sentences to the discussion section.

  • Carbapenem-resistant gram-negative organisms, including CRE, CR Pseudomonas aeruginosa (CRPA), and CR Acinetobacter baumannii (CRAB), pose a major global threat in the 21st century [15, 16].
  • In another Korean study dealing with CRPA and CRAB BSI, carbapenem resistance was an independent risk factor for treatment failure regardless of the appropriateness of empiric antibiotics [31].
  • In a study of CRPA infections by Reyes et al., patients with CP-CRPA infections had higher 30-day mortality than those with non-CP-CRPA infections [33].

Point 4: In discussion comments on treatment:

Savoldi A. BMC Infect Dis 2021; 21:545.

Paul M. Clin Microbiol Infect 2022; 28:521. ESCMID guideline

Kadri SS. Clin Infect Dis 2018; 67:1803.

Response 4: Following the reveiwer’s recommendation, we have inserted the references as references [28], [30], and [27], respectively. Based on these references, we have added the following sentences to the discussion section.

  • Clinicians are prescribing a wide range of treatment regimens in their current practice with no proven efficacy and potential side effects, as available studies do not provide clear recommendations on the type and number of agents to be used in combination [28].
  • Furthermore, the recommended antibiotics for treating KPC-producing CP-CRE BSI such as ceftazidime-avibactam or cefiderocol [29, 30].
  • The presence of difficult-to-treat resistance, defined as resistance to all first-line antibiotics in Gram-negative BSI, is associated with decreased survival rates because of the challenge of finding an effective antibiotic to treat it [27].

Point 5: In conclusion comment on:

Song KH. J Glob Antimicrob Resist 2022; 31:379.

Response 5: We have cited this article in the introduction section as reference [7], where we discuss the current state of research on the topic. We have also mentioned the economic burden associated with antimicrobial resistance in the conclusion section, and have highlighted the need for further research in this area.

  • (Intoduction) In Korea, multidrug-resistant organisms, including Enterobacteriaceae, pose a tremendous clinical and economic burden [6, 7].
  • (Conclusion) Therefore, infection control efforts to prevent CRE acquisition would be a more cost-effective approach to reducing patient mortality.

Point 6: Some typos in introduction: Klebsiella pneumonia should be Klebsiella pneumoniae.

Response 6: We corrected the typos in the manuscript by changing “Klebsiella pneumonia” to “Klebsiella pneumoniae” as suggested by the reviewer.

**We have had the manuscript professionally proofread to ensure that it meets the highest standards of English usage. We are confident that the revised version has addressed theses issues and will be more readable to readers.

We would also like to mention that all the revisions and changes made to the manuscript have been clearly marked and indicated to make it easier for the reviewers to identify. We have used a different font color to distinguish the revised portions from the original text.

Reviewer 4 Report

1、What is the significance of comparing 14-day mortality with 30-day mortality?

2、The number of references is too small. It may also be good to add a reference on the scarcity of information regarding infection control of CRE. One studies reported this to be just over two years.

3、Should include statistics on hospital days and invasive procedure.

4、The significance of the results was not sufficiently discussed.

Minor editing of English language required

Author Response

--------------------------------------------------------------------

Point 1: What is the significance of comparing 14-day mortality with 30-day mortality?

Response 1: We agree that 30-day mortality is a commonly used outcome measure in studies of BSI. However, in our study, we chose to focus on sepsis-attributable mortality, which we believe is a more clinically relevant outcome for the earlier stages of the infection. Therefore, we believe that analyzing both 14-day and 30-day mortality provides a more comprehensive understanding of the impact of CRE BSI on patient outcomes.

Point 2: The number of references is too small. It may also be good to add a reference on the scarcity of information regarding infection control of CRE. One studies reported this to be just over two years.

Response 2: As a result of the revisions suggested by other reviewers and your comment, we have added more than 10 references to our manuscript, which we believe has significantly strengthened the paper. We hope that our revisions have addressed your concerns.

Point 3: Should include statistics on hospital days and invasive procedure.

Response 3: In response to your suggestion, we have added additional information in Table 1 to provide data on foley catheter, central venous catheter, ventilator care, and drainage catheter use in both the CP-CRE and non-CP-CRE group. However, we did not find any significant differences between the two groups in theses parameters.

Point 4: The significance of the results was not sufficiently discussed.

Response 4: Based on the suggestions by reveiwers, we have made signficant changes to our manuscript. We have added several references to strengthen our paper and included a discussion on the finding of our results. We hope that our revisions have addressed your concerns.

** We have had the manuscript professionally proofread to ensure that it meets the highest standards of English usage. We are confident that the revised version has addressed theses issues and will be more readable to readers.

We would also like to mention that all the revisions and changes made to the manuscript have been clearly marked and indicated to make it easier for the reviewers to identify. We have used a different font color to distinguish the revised portions from the original text.

Round 2

Reviewer 4 Report

 It may also be good to add a reference on the scarcity of information regarding infection control of CRE.    

https://doi.org/10.1186/s13756-020-00757-y "Successful control of the first carbapenem- resistant Klebsiella pneumoniae outbreak in

a Chinese hospital 20172019"

Author Response

As the reviewer recommended, we added a reference in the introduction section. 

-> However, there is a lack of studies comparing the characteristics and outcomes between CP-CRE BSI and non-CP-CRE BSI. Furthermore, there is a scarcity of information about management strategies and antibiotic treatment [12]